# Changes in Cortical Activation by Transcranial Magnetic Stimulation Due to Coil Rotation Are Not Attributable to Cranial Muscle Activation

**DOI:** 10.3390/brainsci14040332

**Published:** 2024-03-29

**Authors:** Marco Mancuso, Alessandro Cruciani, Valerio Sveva, Elias Casula, Katlyn E. Brown, Vincenzo Di Lazzaro, John C. Rothwell, Lorenzo Rocchi

**Affiliations:** 1Department of Human Neuroscience, University of Rome “Sapienza”, Viale dell’Università 30, 00185 Rome, Italy; marco.mancuso@uniroma1.it; 2Unit of Neurology, Neurophysiology, Neurobiology, and Psychiatry, Department of Medicine and Surgery, Università Campus Bio-Medico di Roma, Via Alvaro del Portillo 21, 00128 Rome, Italy; alessandro.cruciani@unicampus.it (A.C.); v.dilazzaro@unicampus.it (V.D.L.); 3Fondazione Policlinico Universitario Campus Bio-Medico, Via Alvaro del Portillo 200, 00128 Rome, Italy; 4Department of Anatomical and Histological Sciences, Legal Medicine and Orthopedics, University of Rome “Sapienza”, Piazzale Aldo Moro 5, 00185 Rome, Italy; valerio.sveva@uniroma1.it; 5Department of System Medicine, “Tor Vergata” University of Rome, Via Montpellier 1, 00133 Rome, Italy; elias.casula@gmail.com; 6Department of Kinesiology, University of Waterloo, 200 University Ave W, Waterloo, ON N2L 3G5, Canada; kate.brown@uwaterloo.ca; 7Department of Clinical and Movement Neurosciences, UCL Queen Square Institute of Neurology, University College London, London WC1N 3BG, UK; j.rothwell@ucl.ac.uk; 8Department of Medical Sciences and Public Health, University of Cagliari, Cittadella Universitaria di Monserrato, Blocco I S.S. 554 bivio per Sestu, Monserrato, 09042 Cagliari, Italy

**Keywords:** TMS-EEG, muscle artifact, M1

## Abstract

Transcranial magnetic stimulation coupled with electroencephalography (TMS-EEG) allows for the study of brain dynamics in health and disease. Cranial muscle activation can decrease the interpretability of TMS-EEG signals by masking genuine EEG responses and increasing the reliance on preprocessing methods but can be at least partly prevented by coil rotation coupled with the online monitoring of signals; however, the extent to which changing coil rotation may affect TMS-EEG signals is not fully understood. Our objective was to compare TMS-EEG data obtained with an optimal coil rotation to induce motor evoked potentials (M1_standard_) while rotating the coil to minimize cranial muscle activation (M1_emg_). TMS-evoked potentials (TEPs), TMS-related spectral perturbation (TRSP), and intertrial phase clustering (ITPC) were calculated in both conditions using two different preprocessing pipelines based on independent component analysis (ICA) or signal-space projection with source-informed reconstruction (SSP-SIR). Comparisons were performed with cluster-based correction. The concordance correlation coefficient was computed to measure the similarity between M1_standard_ and M1_emg_ TMS-EEG signals. TEPs, TRSP, and ITPC were significantly larger in M1_standard_ than in M1_emg_ conditions; a lower CCC than expected was also found. These results were similar across the preprocessing pipelines. While rotating the coil may be advantageous to reduce cranial muscle activation, it may result in changes in TMS-EEG signals; therefore, this solution should be tailored to the specific experimental context.

## 1. Introduction

In the past decades, transcranial magnetic stimulation (TMS) has been used extensively to explore cortical excitability in healthy subjects and to investigate the pathophysiology of neurological and psychiatric conditions [1,2,3]. Initial TMS studies have focused on the stimulation of the primary motor cortex (M1) using motor evoked potentials (MEPs) recorded via surface electromyography (EMG) as a readout. In recent years, coupling between TMS and electroencephalography (TMS-EEG) has expanded TMS applications in healthy subjects and patients with neuropsychiatric disorders [4,5,6] due to its ability to provide a functional readout from areas outside M1 and to assess different reactivity and connectivity measures in the time and time/frequency domain [7,8,9,10,11].

One of the main technical issues of TMS-EEG is the activation of cranial muscle by TMS; this may contaminate TMS-EEG responses either directly, as it can induce an early, TMS-locked EMG signal [12,13,14,15], or indirectly, due to somatosensory input caused by muscle twitches [16]. A possible way to address this problem is to remove the EMG signal offline with blind source separation techniques, such as independent component analysis (ICA) or signal-space projection with source-informed reconstruction (SSP-SIR) [13,17,18]. While effective, these methods carry the risk of partly removing EEG signals generated by direct cortical activation by TMS [7,19,20]. Another possibility is to monitor the transcranial evoked potential (TEP) in real time and to adjust the coil rotation to minimize craniofacial EMG [21]. However, a potential drawback of this second approach is represented by the sensitivity of TMS-EEG responses to the direction of current induced in the brain; this effect has been investigated in M1 and other brain areas [22,23,24,25] and may be at least partly linked to the fact that cortical neurons and their inputs show preferential activation for a given current direction [26,27,28,29]. In the aforementioned studies, however, the coil orientations were determined a priori, without online or offline control for cranial EMG activation; it is, therefore, unclear whether the observed changes in TMS-EEG responses found by rotating the coil were caused by differences in cranial muscle twitches.

The aim of the present study was to investigate this issue by comparing TMS-EEG signals, both in the time and time/frequency domain, obtained under two conditions: (1) M1 stimulation with the coil handle pointing 45° backwards, to induce posterior–anterior (PA) current in the brain, known to be optimal for the stimulation of this area [26,27,28,29] and (2) M1 stimulation performed during the online monitoring of TEPs, with the aim of minimizing cranial EMG activation. The data were also analyzed with two different approaches to remove cranial EMG activation (ICA and SSP-SIR), to ensure that the results were not biased by the preprocessing pipeline, as previously suggested [19].

## 2. Materials and Methods

### 2.1. Participants

Sixteen healthy right-handed subjects [30] (age 31.63 ± 3.96, ten males, six females) participated in the study. They had no history of neurological or psychiatric disorders and were not taking drugs active in the central nervous system. All the procedures were performed in accordance with the Declaration of Helsinki and approved by the review board of University College London (ID 10037/001, approval date 22 January 2016). Participants gave written informed consent prior to the experimental session.

### 2.2. Experimental Design

Experiments were conducted in a quiet room, with subjects sitting in comfortable chairs, with their forearms resting on a pillow on their lap. We asked them to fixate on a white cross (6 × 6 cm) in front of them to avoid eye movements during the experiment. A masking noise was continuously played through earphones worn by the subjects, which was composed of white noise mixed with specific time-varying frequencies of the TMS click with the aim of minimizing the auditory evoked potential (AEP) caused by the latter [31]. The volume of the masking noise was calibrated for each participant, being gradually raised until the participant confirmed that the click produced by the TMS was inaudible, or until the participant could no longer comfortably endure the noise (always below 90 dB) [32]. Also, to minimize the influence of the TMS click on EEG recordings, ear protectors (SNR = 30) were positioned over the earphones. This method is validated for its efficacy in diminishing the AEP triggered by subthreshold TMS [5,33]. Additionally, a 0.5 cm foam layer was placed underneath the coil to minimize the bone conduction of the TMS click [33,34,35]. Both the stimulation hotspot and RMT were found with subjects wearing EEG caps and the described foam layer. 

Each subject completed a single recording session where two blocks of 150 TMS stimuli over the left M1 were delivered (inter-pulse interval 2 s, jitter 10%). In the first block, the coil handle pointed backwards approximately 45° away from the midline, inducing a current in the cortex in the PA direction (M1_standard_) [1,29]. In the second block, the stimulated cortical hotspot was the same, but EEG responses to TMS were monitored online using previously published methods [21] and the orientation and tilting of the coil were modified to minimize cranial EMG activity (M1_emg_). At the end of each block, subjects were asked to score the loudness of the TMS click with a visual analogue scale (VAS) ranging from 0 (no perception) to 10 (maximal perception). The same assessment was conducted for possible discomfort caused by TMS (0 = no discomfort; 10 = maximal discomfort).

### 2.3. Transcranial Magnetic Stimulation and Electromyography

Monophasic pulses were administered using a 70 mm figure-of-eight coil linked to a Magpro X100 magnetic stimulator (MagVenture, Farum, Denmark) for TMS procedures. The “hotspot” of the left M1 was located by moving the coil to find the spot where the largest MEP was elicited in the right first dorsal interosseous (FDI) muscle. The stimulation site was identified in MNI space, and the coil’s position was kept stable during the entire stimulation using a Brainsight neuronavigation system (Rogue Research Inc., Montreal, QC, Canada) in combination with a Polaris Spectra optical tracking system (Northern Digital Inc., Waterloo, ON, Canada). An estimated MRI scan in MNI space was utilized for each subject. The accuracy of the coil’s position during each TMS session was recorded in MNI coordinates to assess any deviation in positioning (linear, angular, and twist deviations). The intensity of the stimulation was set to 90% of the resting motor threshold (RMT), defined as the lowest intensity at which MEPs with a minimum amplitude of 50 μV are produced in approximately half of ten trials [1]. EMG responses from the FDI muscle (belly-tendon montage) were captured with 10 mm Ag/AgCl cup electrodes, digitized at a 5 kHz sampling rate using a CED 1401 A/D laboratory interface (Cambridge Electronic Design, Cambridge, UK), then amplified (with a gain of 1000×) and band-pass filtered (from 5 Hz to 2 kHz) using a Digitimer D360 amplifier (Digitimer Ltd., Welwyn Garden City, Hertfordshire, UK).

### 2.4. Electroencephalographic Signal Recording and Analysis

We recorded EEG with a DC-coupled TMS-compatible amplifier (Brainamp DC, Brain Products GmbH, Gilching, Germany). Each subject underwent scalp preparation using an abrasive/conductive gel (V17 Abralyt 2000, Brain Products GmbH, Gilching, Germany). The EEG was digitized at a 5 kHz sampling rate from the following passive electrodes (Easycap GmbH, Wörthsee, Germany): FP1, FPz, FP2, AF7, AF3, AF4, AF8, F7, F5, F3, F1, Fz, F2, F4, F6, F8, FT7, FC5, FC3, FC1, FC2, FC4, FC6, FT8, T7, C5, C3, C1, Cz, C2, C4, C6, T8, TP9, TP7, CP5, CP3, CP1, CPz, CP2, CP4, CP6, TP8, TP10, P7, P5, P3, P1, Pz, P2, P4, P6, P8, PO7, PO3, POz, PO4, PO8, O1, Oz, O2, FCz, and AFz. Two further electrodes were placed on the forehead as the online reference and ground. Impedances were kept lower than 5 kΩ.

We preprocessed EEG offline with EEGLAB 14.1.1 [36], and some functions included in the TMS-EEG signal analyzer (TESA) toolbox [37], all running in MATLAB environment (Version 2018b, MathWorks Inc., Natick, MA, USA). The EEG signal recorded during TMS was epoched (−1.3 to 1.3 s) using a baseline from −1000 to −10 ms and the TMS artifact was removed from −5 to 12 ms around the pulse and excessively noisy EEG epochs were excluded after the visual inspection of the data. After epoching, the early, TMS-locked artifacts [12] were removed by means of a fastICA algorithm; only the 15 components explaining the largest variance were plotted in a time window ranging from −200 to 500 ms, and only those reflecting residual scalp EMG or voltage decay were eliminated after visual inspection, based on time, frequency, scalp distribution, and amplitude criteria [13]. Subsequently, the missing data were interpolated with a cubic interpolation; the signals were down-sampled to 1000 Hz, and the bandpass (1–100 Hz) and bandstop (48–52 Hz) were filtered with a fourth-order Butterworth filter. The epochs were restricted to between −1 and 1 s to reduce possible edge artifacts. The data around the TMS pulse were again removed as before, and a second round of fastICA was then performed, plotting the full epoch length, to remove residual artifacts non-time-locked with the TMS pulse (e.g., spontaneous eye blinks and continuous muscle activity). Lastly, a final cubic interpolation was applied, and the EEG signals were re-referenced to the common average reference. This preprocessing pipeline will be referred to as ICA-ICA.

As a consensus regarding the best preprocessing method for TMS-EEG is lacking [7], we also used an alternative solution to remove the early, TMS-locked artifacts, i.e., SSP-SIR [17], which replaced the first round of ICA. The second pipeline will be referred to as SSP-SIR-ICA (see Table 1).

### 2.5. Data Analysis and Statistics

The following variables were compared for each stimulated area in M1_standard_ and M1_emg_ recording conditions: MNI coordinates (x, y, z), errors in coil positioning (linear, angular, twist), and VAS scores for residual auditory sensation and discomfort. Because most of the variables were non-normally distributed (*p*-values of the Shapiro–Wilk test < 0.05) and due to the small sample size, the aforementioned comparisons were performed by means of the Wilcoxon signed-rank test using SPSS version 25 (SPSS Inc., Chicago, IL, USA). Also, the change in coil rotation between M1_standard_ vs. M1_emg_ was recorded and reported.

The EEG signals were analyzed using scripts written in Matlab, version 2018b (MathWorks Inc., Natick, MA, USA) and using functions from the EEGlab and Fieldtrip toolbox [38]. First, we investigated cranial EMG activation in the two conditions by comparing the average amplitude of rectified activity in a 5–25 ms interval and recorded the electrode showing the largest muscle artifact via a Wilcoxon signed-rank test. Following this, we performed an analysis whose general aim was to investigate differences and similarities of TEPs, and oscillations obtained in M1_standard_ and M1_emg_ conditions, considering the two preprocessing pipelines used to remove early, TMS-locked artifacts separately (ICA-ICA and SSP-SIR-ICA). First, we assessed possible differences in EEG amplitude. To do this, the TEPs were first divided into discrete time windows (TWs), identified based on the grand average (GA) signals in all subjects and both conditions, in a region of interest (ROI) including electrodes close to the stimulation site (F3, FC3, FC1, F1, C3, C1, Fz, FCz, and Cz). Time windows of interest (TWs) were chosen around peaks of activity in the GA TEPs, using the minima between peaks as boundaries. The average TEP amplitude was calculated in each TW and values from each electrode were compared between recording sessions by means of paired *t*-tests, with cluster-based permutation corrections for multiple comparisons (5000 permutations).

In a second analysis, we sought to investigate similarities between the signals recorded in M1_standard_ and M1_emg_ conditions. To do so, we calculated the concordance correlation coefficient (CCC) between TEPs in each time window (obtained as previously explained). The CCC is a form of intraclass correlation coefficient that takes both the covariance and the absolute distance between two distributions into account. It is calculated as follows:$$\mathrm{CCC}=\frac{{2\sigma }_{12}}{\sigma_{21}+\sigma_{22}+{\left({µ}_{1}-{µ}_{2}\right)}^{2}}$$
where *σ*_12_ is the covariance between two distributions, *σ*_2*x*_ is the variance of distribution x, and *µx* is the average of distribution x [39]; this method has been previously used in the TMS-EEG setting [14,18,33,40]. This statistical analysis entailed an extreme-corrected permutation approach, where the CCC was first calculated between conditions in each electrode and time window; after this, a custom distribution of CCC values was built by shuffling each subject’s trials between the two conditions (M1_standard_, M1_emg_) 1000 times. The CCC values found were deemed significant if their values fell in the lower fifth percentile of the custom distribution, divided by the number of TWs to account for type I errors across multiple observations. We then applied an extreme correction to solve the problem of multiple comparisons across electrodes. To this end, we built a distribution using the lowest CCC values found among the 63 electrodes in each permutation, and only CCC values that fell within the lowest fifth percentile of this distribution, divided again by the number of TWs, were considered significant [41]. This resulted in one-tailed statistics aimed at identifying electrodes that show a CCC value smaller than that expected under the null hypothesis (i.e., assuming that the two sessions were performed with the same coil position). Lastly, we performed Morlet wavelet-based decomposition (3 to 10 cycles, logarithmically increasing from 1 to 40 Hz), which was used to calculate TMS-related spectral perturbation (TRSP) and intertrial phase clustering (ITPC) according to the previous literature [42,43]. While standard frequency ranges of interest were used (delta 1–4 Hz, theta 5–9 Hz, alpha 9–12 Hz, beta 13–20 Hz, and gamma 21–40 Hz) [11,43], we identified time ranges for each frequency using a data-driven approach. We collapsed M1_standard_ and M1_emg_ TRSP and ITPC data from all subjects into one grand average for each preprocessing pipeline. TRSP data were z-corrected over each frequency to a baseline between −500 ms and −200 ms before the TMS pulse. Grand average TRSP and ITPC data were z-transformed with respect to all values in the time/frequency analysis and all z-values below 1.64 were excluded (roughly corresponding to an uncorrected *p*-value of 0.05). Three broad signal clusters were identified, with frequencies ranging from 3 to 8 Hz (delta and theta), 9 to 20 Hz, (alpha and beta), and 21 to 40 Hz (beta and gamma). Time ranges were selected accordingly for each frequency range of interest. TRSP and ITPC values from each subject were then averaged according to condition (M1_standard_ and M1_emg_) in the pre-selected frequency bands. As for TEPs, values in all electrodes were compared pairwise via *t*-tests, and a cluster-based permutation correction (5000 permutations) was applied to correct for multiple comparisons.

### 2.6. Data and Toolboxes Availability Statement

All used toolboxes (EEGlab and fieldtrip) are publicly available. The data are available upon the provision of a formal data-sharing agreement.

## 3. Results

The test sessions were tolerated by all subjects and no adverse effects were reported. M1_standard_ and M1_emg_ did not significantly differ in terms of the location of stimulated areas, precision in coil positioning, and VAS scores (Table 2). Interindividual differences in coil angle are presented in Table 3.

To minimize cranial muscle activation, in the M1_emg_ condition, the coil handle was consistently rotated posteriorly, so as to get an angle difference of nearly 40° with respect to the M1_standard_ condition (Table 3).

As expected, the early, TMS-locked artifact caused by cranial muscle activation was considerably lower in amplitude in the M1_emg_ than in the M1_standard_ condition (Figure 1, *p* = 0.024, median M1_emg_ = 23.93 µV, median M1_standard_ = 3.83 µV).

For TEP analysis, six similar TWs were identified for the ICA-ICA pipeline (TW1: 11–22 ms; TW2: 23–41 ms; TW3: 42–58 ms; TW4: 59–75 ms; TW5: 76–108 ms; and TW6: 109–180 ms) and SSP-ICA (TW1: 11–17 ms; TW2: 18–41 ms; TW3: 42–58 ms; TW4: 59–75 ms; TW5: 76–113 ms; and TW6: 113–180 ms); see Figure 2.

With the ICA-ICA pipeline (number of rejected components for M1_standard_ = 12.3 ± 2.3, and for M1_emg_ = 11.8 ± 3.2), a more pronounced local negativity was found in M1_standard_ in TW1, corresponding to the N15, together with a more negative N45 right-hemispheric component (TW3) and a more pronounced N100 (TW5) with respect to M1_emg_ (Figure 3). A similar pattern was identified with the SSP-SIR-ICA pipeline (number of rejected components for M1_standard_ = 11.3 ± 2.1, and for M1_emg_ = 10.2 ± 2.3), albeit with a more restricted set of results, with statistically significant differences only in TW3 and TW5, corresponding to the N45/P60 and N100 TEP components, respectively (Figure 4). The CCC analysis yielded comparable results, with lower-than-expected values obtained in most TWs with both the ICA-ICA (TW1, TW2, TW4, TW5) and the SSP-ICA (TW1, TW2, TW5, TW6) pipelines (Figure 3 and Figure 4, respectively). The TWs identified for time/frequency analysis were 0–250 ms for delta and theta, 0–160 ms for alpha and beta, and 0–100 ms for gamma bands (Figure 5 and Figure 6). The TRSP analysis showed greater M1_standard_ power in theta, alpha, and beta power when the ICA-ICA pipeline was considered; results in the same direction were confirmed by the SSP-SIR-ICA pipeline in the alpha and gamma power (Figure 7). ITPC followed a similar pattern, with larger values in the M1_standard_ condition in all frequency bands when using the ICA-ICA pipeline and larger values in theta, alpha, and beta bands with the SSP-SIR-ICA pipeline (Figure 8).

## 4. Discussion

In this study, we compared TMS-EEG signals from M1 stimulation using either a PA current orientation, which maximizes descending output from M1 and is often chosen for TEPs [25,43,44], or an orientation aimed at maximally reducing cranial EMG. The purpose of this comparison was to investigate whether coil rotation changes TMS-EEG responses while carefully controlling for local muscle activation. We also used two different preprocessing pipelines to exclude potential sources of bias due to the suboptimal recovery of neural sources of TEPs. While different methods, such as ICA and SSP-SIR, are available [13,17,45], their use may lead to different results [19]. Also, there is no general consensus on which method is optimal to suppress the early EMG activity which is time-locked to the TMS pulse [7].

Regardless of the algorithm used, our results indicate that M1 TMS-EEG signals obtained with a PA induced current direction are larger than those following coil rotation aimed at minimizing cranial EMG activity, a result not attributable to differences in coil position (Table 2). This was evident when comparing time-domain signals (TEPs) and affected several of the main TEP peaks, including the N15, N45, P60, and N100. TEPs are thought to reflect a summation of excitatory and inhibitory postsynaptic potentials generated by the activity of cortical pyramidal neurons and interneurons [32,46]. Early M1 TEPs (<40 ms) are recorded as radial EEG dipoles and probably reflect periods of increased cortical neuronal discharge [47,48]; they are sensitive to the effects of sodium and potassium channel blockers [49,50] and may be generated by the interplay of cortex and subcortical sources, such as the basal ganglia and thalamus [5,51]. Neural processes contributing to later TEP components are likely different, with the N45/P60 constituting a tangential dipole where the positive peak is caudal to the stimulation site and is decreased by glutamatergic AMPA receptor blockers [52], while the negative wave is located in the frontal regions contralateral to the stimulated hemisphere and is possibly linked with GABAa transmission [53]. Lastly, the N100 is a later component with maximal amplitude at the stimulated cortex [18,33], thought to represent at least partially the activation of GABAb receptors [53,54].

The results obtained with TEPs were confirmed by the analyses of TMS-induced oscillations. The topographical distribution of activity in all frequency bands was similar between M1_standard_ and M1_emg_ conditions, with delta and theta ERSP and ITPC widely distributed on the scalp; alpha and beta showing a local maximum with a tendency to spread to contralateral frontal areas; and gamma keeping an activation profile more limited to the stimulation site. Differences between conditions were clearer in the alpha, theta, and beta bands, particularly at the stimulation site and contralateral hemisphere, with a sparing of the vertex area. The physiological underpinnings of TMS-induced EEG oscillations are less characterized compared to TEPs, with some pieces of evidence suggesting that oscillations in the beta and alpha bands may involve the activation of local populations of inhibitory cortical neurons [55,56].

Overall, our results suggest that modifying the orientation of the TMS coil in an attempt to minimize cranial EMG activity changes the amplitude of most TEP components, as well as power and phase consistency across trials of EEG signals in several frequency bands up to 50 Hz. This conclusion is supported by the comparisons performed between the two experimental conditions and by the CCC analysis, which suggests a similarity between signals collected with M1_standard_ and M1_emg_ coil orientation that is significantly lower than that under the null hypothesis in most TEP components. Considering the discussion above, it is likely that the TEPs and TMS-induced oscillations explored in the present study are generated by diverse physiological mechanisms; therefore, it is reasonable to conclude that a specific sensitivity to a TMS current direction may not be confined only to inputs to corticospinal tract neurons [27,29], but may also be shared by neural elements with very different physiological properties contributing to the generation of TMS-EEG signals.

One important question is whether the observed differences between M1_standard_ and M1_emg_ stimulations might be due to artifacts or sensory EEG responses. In this work, we compared two conditions characterized by different cranial EMG activation (i.e., larger in M1_standard_ and smaller in M1_emg_, as depicted in Figure 1). Although this early artifact was removed by means of ICA or SSP-SIR, according to the pipeline used, it is worth noting that these methods do not ensure a complete separation between signals generated by neural sources and artifacts [7]. Therefore, despite the fact the scalp distribution of the N15 in the present work does not correspond to that of cranial EMG activation following M1 stimulation, the latter being more lateral and presenting as a local dipole [13], some uncertainty remains about the differences in early signals between the two conditions. However, given that the EMG artifact lasts on average less than 30 ms (Figure 1), it cannot justify the differences observed in later signals, especially in the 100–200 ms range. Previous work has indicated that TMS-EEG responses in this time window can be contaminated by potentials evoked by auditory and somatosensory stimulation [33,57,58]. It is unlikely that auditory evoked potentials (AEPs) have skewed our results for several reasons, the first being the masking of auditory input by a continuous noise and a foam layer beneath the coil [31,33]. The effective suppression of AEP is indicated by the absence of a prominent vertex N100/P200, which often indicates the imperfect masking of the TMS click [33], and by the low auditory VAS scores between the two conditions explored. The fact that M1_standard_ and M1_emg_ recording blocks shared the same stimulation intensity and, thus, a similar loudness of the TMS click, as confirmed by the lack of significant differences in the VAS scores, further supports the notion that our results are not due to contamination by AEPs. A possible overlap between somatosensory evoked responses (SEPs) and TEPs may represent a more pertinent problem in the present experimental setting. Since it has been suggested that afferent input due to cranial muscle contraction may induce measurable EEG responses [16], it is conceivable that the larger muscle twitch in the M1_standard_ condition may have driven the observed differences in TMS-induced signals. However, it has been suggested that somatosensory input due to cranial muscle twitches induces a multimodal vertex N100/P200, similar to AEPs, rather than a lateralized signal compatible with the modality-specific activation of the somatosensory cortex [16,59]. By contrast, the topography of significant differences between M1_standard_ and M1_emg_, both in the time and time/frequency domains, is lateralized and, thus, likely driven by a different direct cortical activation due to TMS. This is further indirectly supported by the similar VAS scores for discomfort in the two conditions, which is highly correlated with cranial muscle twitches [16,60]; this result probably indicates that the difference in muscle activation was not large enough to generate a different percept and, by extension, a different multimodal vertex potential due to saliency of stimulation [33]. Finally, it is interesting to note that the two processing pipelines did not produce exactly the same effects on the data, indicating that they are not equivalent. In the absence of any “ground truth” in the present experiments, it is not possible to draw any firm conclusions about the reasons for the differences. This could perhaps be addressed in future experiments on manufactured datasets that include different types of artifacts [61].

## 5. Conclusions

In conclusion, the present results suggest that coil rotation influences TMS-evoked EEG potentials and oscillations, a result not attributable to cranial muscle activation and independent of the preprocessing pipeline employed to suppress early EMG artifacts. This finding is of unclear physiological significance, as only electrophysiological variables were investigated. However, TEP changes have been demonstrated following sensorimotor integration tasks [14,15,44] and in neurological disorders [4,5,62]; therefore, future studies are warranted to understand whether changes in TMS-EEG responses due to coil rotation also affect results in the context of behavioral experiments or clinical populations. If this were the case, the technical advantage represented by the reduction in the EMG artifact should be balanced, in the specific experimental context, against a potential loss of physiological information.

## Figures and Tables

**Figure 1 brainsci-14-00332-f001:**
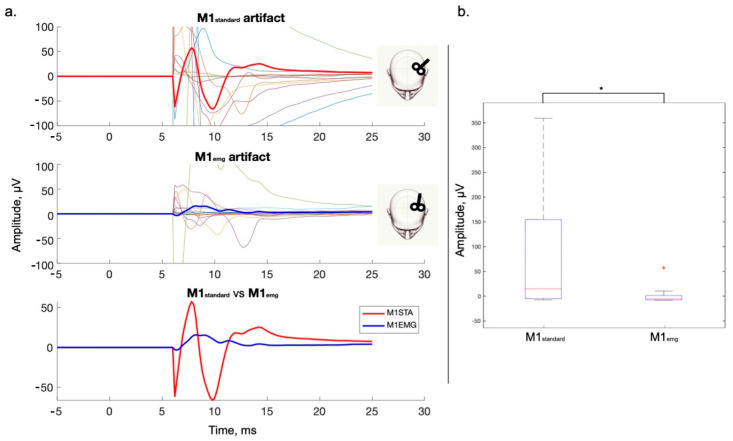
Muscle artifact comparison between sessions, Fc5 electrode. (**a**) The unpreprocessed signal is depicted in a time window compatible with muscle artifact occurrence (5–25 ms). Fc5 was chosen for representation as it showed the greatest grand average signal in this time window. The signal before 6 ms is zeroed out to reduce the visual impact of the TMS stimulus artifact. No formal comparison was made due to the great variability in muscle anatomy in each subject. **Upper row**: muscle artifact from M1_standard_; **middle row**: muscle artifact from M1_emg_; **lower row**: M1_standard_ and M1_emg_ artifact. (**b**) Boxplots representing the averaged rectified Fc5 signal from each condition between 5 and 25 ms. Wilcoxon signed-rank test showed a significant difference between the two conditions. +: datapoint beyond2 SD from the mean; star: significant difference.

**Figure 2 brainsci-14-00332-f002:**
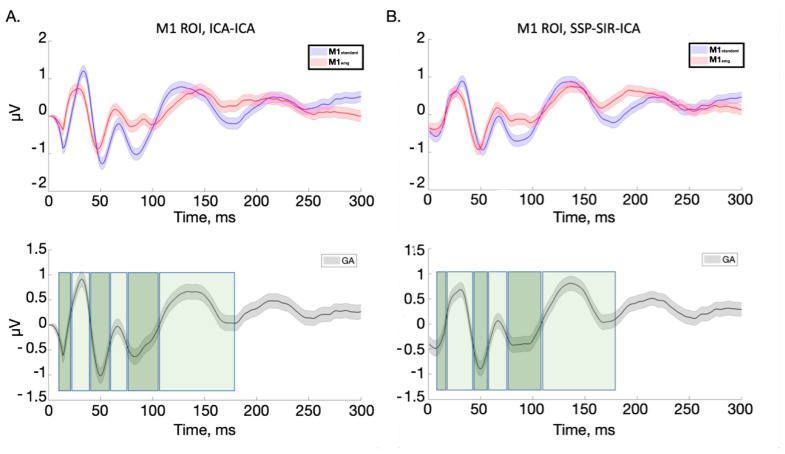
TEPs averaged from electrodes close to the stimulation site (F3, FC3, FC1, F1, C3, C1, Fz, FCz, and Cz). The upper row of panel (**A**) depicts signals obtained with the ICA-ICA pipeline, with red and blue lines, respectively, indicating the M1_standard_ and M1_emg_ conditions; in the lower row, the grand average signal across the two conditions is represented, with green boxes indicating the time windows used for further analysis. Panel (**B**) shows the same variables obtained with the SSP-SIR-ICA preprocessing pipeline. Shaded areas indicate the standard error of the mean.

**Figure 3 brainsci-14-00332-f003:**
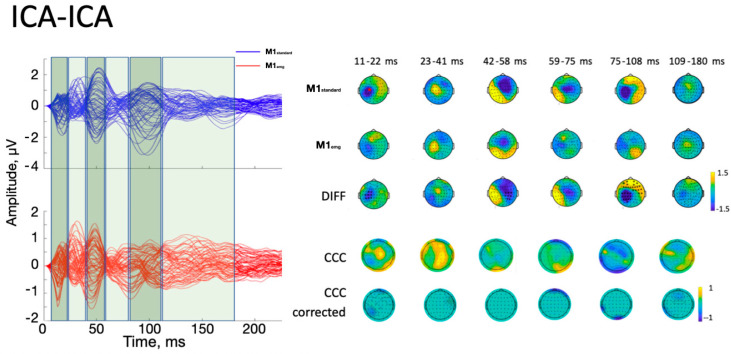
Results of time-domain analyses using the ICA-ICA pipeline. On the left, average butterfly plots of M1_standard_ (blue lines) and M1_emg_ (red lines) are depicted, together with time windows used for statistical analysis (green boxes). Each line represents signals from one electrode. The first three rows of the right part of the figure represent the average signal in each ToI for M1_standard_, M1_emg_, and their difference, respectively. Black asterisks indicate statistically significant differences (*p* < 0.05). The fourth row represents uncorrected CCC values between conditions, while the fifth row indicates lower-than-expected CCC values between conditions after cluster-based correction (*p* < 0.05; see text for details). Red star in the first head model: stimulation point.

**Figure 4 brainsci-14-00332-f004:**
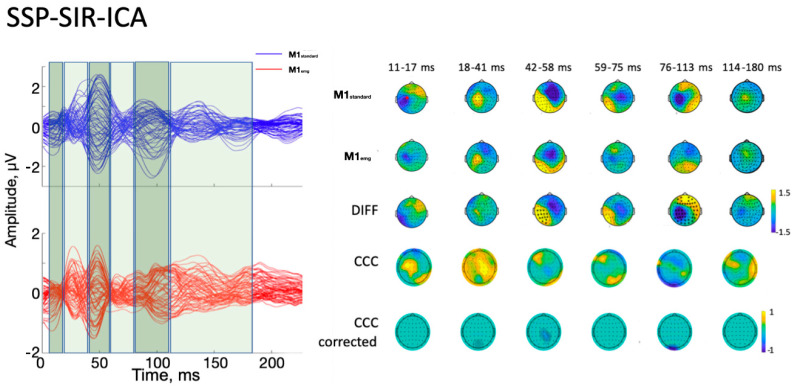
Results of time-domain analyses using the SSP-SIR-ICA pipeline. On the left, average butterfly plots of M1_standard_ (blue lines) and M1_emg_ (red lines) are depicted, together with time windows used for statistical analysis (green boxes). Each line represents signals from one electrode. The first three rows of the right part of the figure represent the average signal in each ToI for M1_standard_, M1_emg,_ and their difference, respectively. Black asterisks indicate statistically significant differences (*p* < 0.05). The fourth row represents uncorrected CCC values between conditions, while the fifth row indicates lower-than-expected CCC values between conditions after cluster-based correction (*p* < 0.05; see text for details). Red star in the first head model: stimulation point.

**Figure 5 brainsci-14-00332-f005:**
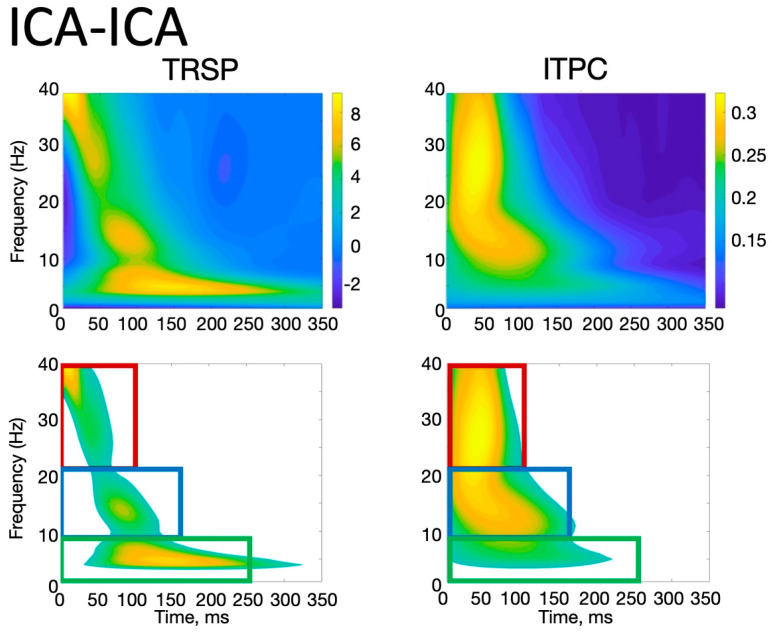
Time/frequency window identification procedure. **Upper row**: TRSP and ITPC obtained by averaging signals in all subjects, conditions, and a cluster of electrodes close to the stimulation site (F3, FC3, FC1, F1, C3, C1, Fz, FCz, and Cz) using the ICA-ICA preprocessing pipeline. The squares in the lower row show the time/frequency regions used to extract TRSP and ITPC values (z = 1.64, see text for details).

**Figure 6 brainsci-14-00332-f006:**
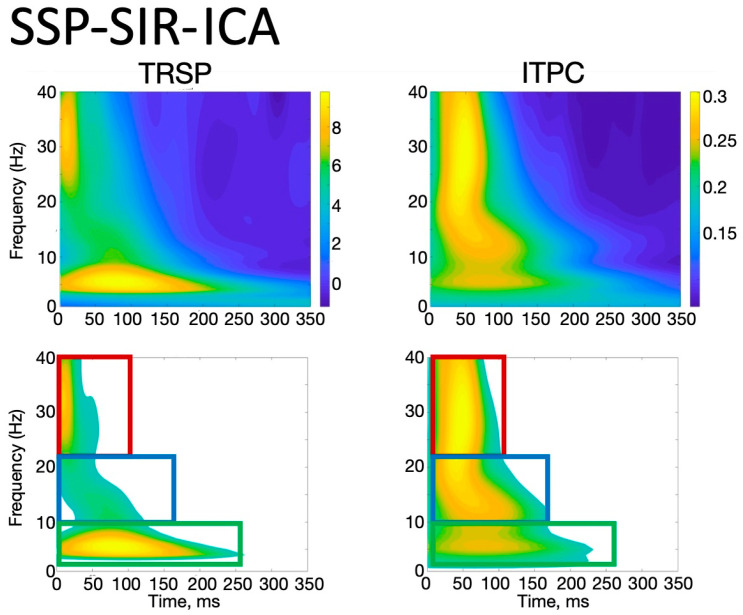
Time/frequency window identification procedure. **Upper row**: TRSP and ITPC obtained by averaging signals in all subjects, conditions, and a cluster of electrodes close to the stimulation site (F3, FC3, FC1, F1, C3, C1, Fz, FCz, and Cz) using the SSP-SIR-ICA preprocessing pipeline. The squares in the lower row show the time/frequency regions used to extract TRSP and ITPC values (z = 1.64, see text for details).

**Figure 7 brainsci-14-00332-f007:**
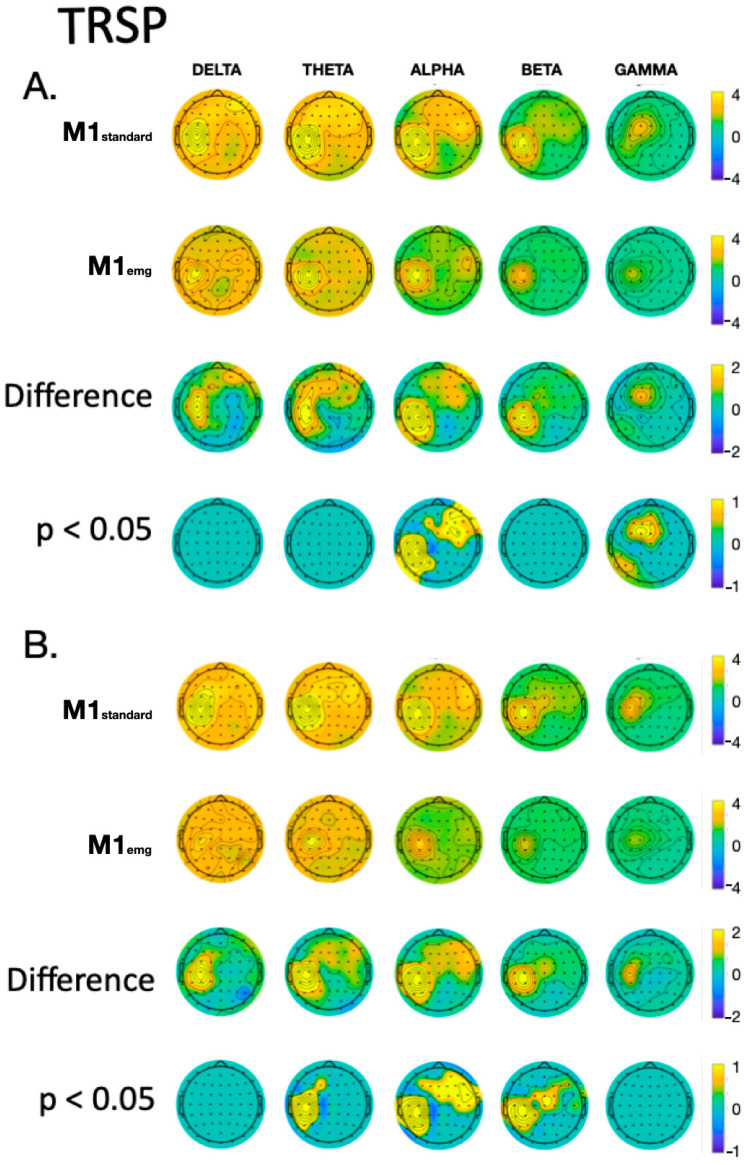
Comparison of TRSP values between M1_standard_ and M1_emg_ conditions in discrete frequency bands, using ICA-ICA (panel (**A**)) and SSP-SIR-ICA (panel (**B**)) pipelines. In each panel, the first two rows give a qualitative representation of TRSP values, the third row depicts the difference between conditions, and the fourth indicates statistically significant differences.

**Figure 8 brainsci-14-00332-f008:**
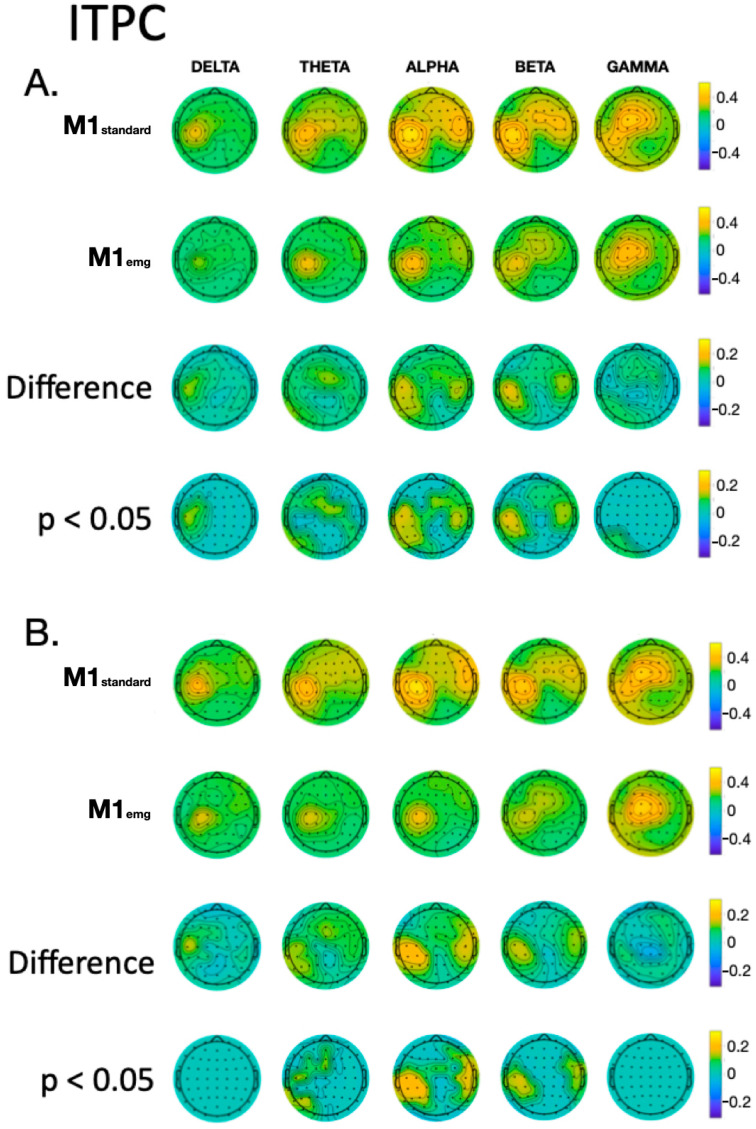
Comparison of ITPC values between M1_standard_ and M1_emg_ conditions in discrete frequency bands, using ICA-ICA (panel (**A**)) and SSP-SIR-ICA (panel (**B**)) pipelines. In each panel, the first two rows give a qualitative representation of TRSP values, the third row depicts the difference between conditions, and the fourth indicates statistically significant differences.

**Table 1 brainsci-14-00332-t001:** Comparison between the two preprocessing pipelines utilized.

	Pipeline 1 (ICA-ICA)	Pipeline 2 (SSP-SIR-ICA)
	Epoching, TMS artifact removal and interpolation, bad epoch removal	
Muscle artifact rejection	ICA	SSP-SIR
	Down-sample, bandpass, and bandstop filtering, epoch restriction	
Other artifactual component (eye blinks, electrode noise, etc.) removal	ICA	ICA
	Average re-referencing	

**Table 2 brainsci-14-00332-t002:** Summary of raw values and between-session comparisons of MNI coordinates, coil position errors, and VAS scores. No difference was found between the conditions for any of the items. VAS: visual analogue scale. MNI coordinates and coil position errors are expressed in mm.

	M1_standard_	M1_emg_	Z	*p*
MNI coordinate (x)	−46.09 ± 0.86	−45.25 ± 0.77	−1.448	0.148
MNI coordinate (y)	−11.16 ± 1.09	−10.94 ± 1.42	−0.931	0.352
MNI coordinate (z)	70.19 ± 1.41	70.06 ± 1.29	−0.595	0.552
Error (linear)	1.58 ± 0.12	1.55 ± 0.12	−0.181	0.856
Error (angular)	2.80 ± 0.24	2.67 ± 0.25	−1.138	0.255
Error (twist)	3.01 ± 1.45	2.89 ± 1.48	−1.533	0.125
VAS score (auditory)	0.25 ± 0.11	0.25 ± 0.11	<0.01	1
VAS score (discomfort)	0.31 ± 0.12	0.25 ± 0.12	−0.09	0.976

**Table 3 brainsci-14-00332-t003:** Single subject-level rotation of the coil from M1_standard_ to reach M1_emg_. All rotations are expressed in degrees, with an anti-clockwise direction (toward the sagittal midline). Notice how all the rotations ranged between 30 and 44°, showing low inter-subject variability.

Anti-Clockwise Rotation (Degrees)
38.45
30.34
40.23
35.43
31.45
39.43
43.42
44.33
39.45
41.34
31.09
41.92
37.99
42.9
43.21
42.99

## Data Availability

All used toolboxes (EEGlab and fieldtrip) are publicly available. The data presented in this study are available upon request from the corresponding author. The data are not publicly available due to privacy restrictions.

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
