# Peer review of "Changes in Cortical Activation by Transcranial Magnetic Stimulation Due to Coil Rotation Are Not Attributable to Cranial Muscle Activation"

_brainsci, 2024, doi:10.3390/brainsci14040332_

Round 1

Reviewer 1 Report

Comments and Suggestions for Authors

Thank you for giving me the opportunity to review your interesting TMS-EEG paper. I have no particular objection to the outline, but please address the following two points.

1.  Many of the TEP traces in the upper and middle panels of Figure 1 are scaled out of the graph, so adjust the Y-axis range to the appropriate scale.

2.  It would be better to explain the difference between M1_STANDARD and M1_EMG conditions more clearly and concretely in the Methods section with an illustration.

Author Response

Thank you for giving me the opportunity to review your interesting TMS-EEG paper. I have no particular objection to the outline, but please address the following two points.

  1. Many of the TEP traces in the upper and middle panels of Figure 1 are scaled out of the graph, so adjust the Y-axis range to the appropriate scale.

Thanks for the comment. It is true that, due to the great variability between subjects in terms of muscle artifact, we had to produce a figure that scales the individual traces out of the field in order to focus on the mean of each condition. Nonetheless, since allowing all traces in the field would flatten the mean and render it uninterpretable, we decided to also represent the data in terms of averaged signal between 5-25 ms in a boxplot (figure 1 ha been modified accordingly). Also, the averaged signal was compared between sessions by means of a Wilcoxon signed rank test (188-190, 263-264). This will provide a clearer outlook on the variability of data between subjects

  1. It would be better to explain the difference between M1_STANDARD and M1_EMG conditions more clearly and concretely in the Methods section with an illustration.

Thanks for the suggestion. Indeed, a representation of the different coil directions in the two conditions is already present in figure 1, and hence serves the role of the figure suggested by the reviewer. Therefore, we fear that another depiction of the methodology might be redundant.

See the attached files for the updated version of the text

Reviewer 2 Report

Comments and Suggestions for Authors

The authors of the manuscript titled "Changes in cortical activation by transcranial magnetic stimulation due to coil rotation are not attributable to cranial muscle activation" are conducting a research study investigating a "blind" vs EEG-Informed TMS open loop approach in the motor area. The topic caught my attention, as I find it intriguing that the optimal coil orientation to evoke an MEP may differ from that to evoke a TEP, owing to technological (EEG) or physiological factors (exogenous or endogenous). I hope the authors will delve deeper into this topic in future investigations.

I have some feedback to offer that I believe could assist the authors in enhancing their article.

Abstract:

·      I would suggest revising the term "confound" in line 24 and "TMS-EEG signals" in line 27.

·      I found it challenging to follow the distinctions among M1_STANDARD, M1_EMG, and M1_EMG TMS-EEG throughout the paper. Perhaps the authors could use a simpler labeling system, such as Experiment 1, Experiment 2, or a similar approach, defined at the outset and consistently applied throughout the manuscript.

·      I recommend that the authors clearly articulate the problem, the significance of muscle activation and why this has triggered your research question, and emphasize the key findings.

Introduction:

·      I would recommend revising the terminology employed in this article. Additionally, careful consideration should be given to the use of "evoke" versus "induce," as well as distinguishing between "muscle artifacts" and "EMG activity."

·      Lines 71-74 are somewhat unclear. Clarifying the statement regarding the drawbacks of monitoring TEP potentials online would be beneficial, as it seems to represent a pivotal aspect of the manuscript's objectives.

·      Bullet points summarizing the contributions would help clarify the authors' focus.

Experimental Design:

·      The use of noise masking to minimize expectations and TMS sound perception is standard practice in TMS. I'm curious about the effects of continuous noise on cortical excitability. Are there any known effects? There is a recently paper based on MEG study on this topic.

·      Were the hotspot and rMT determined with the subject wearing a cap and foam? Please, add this info. 

·      More detail regarding about the meaning of "MEPs with the lowest possible stimulation intensity" in line 113 would be helpful.

Transcranial Magnetic Stimulation and Electromyography:

·      Additional information about the EMG montage used would be beneficial.

·      Including information about the Inter-Pulse Interval (IPI) utilized would enhance clarity.

·      Regarding line 156, was the TMS artifact removed and after the EEG signal interpolated?

·      A table comparing and detailing the first versus second pipeline would be informative.

Statistical Analysis:

·      Did the authors employ any techniques to mitigate the Simpson paradox?

·      Could a linear mixed model be suitable for conducting multiple comparison statistical analyses on multidimensional data?

·      Clarification regarding the term "first analysis" in line 189 is recommended.

·      What assumptions were made before conducting Canonical Correlation analysis?

·      The meaning of "extreme" corrected permutation approach in line 207 requires clarification.

·      Did the authors apply baseline correction before conducting TF analysis?

Results:

·      Table 1 is insightful. Including the formulas used to calculate the mean and standard deviation, it is something I would recommend. Additionally, a more informative caption is suggested. The same applies to Table 2.

·      Figure 1 would benefit from including units on the y-axes. Consider presenting a plot with shaded ranges representing standard deviation to illustrate the overlap of muscle artifact between approaches.

·       How many ICA components were rejected?

·      A group-level topoplot for each condition would provide insight into TEP characteristics.

·       Figures 2 and 3 raise concerns about the range of butterfly plots magnitude relative to the expected motor area range.

·      Investigating MEP amplitude variability between different approaches could yield valuable insights for further statistical analysis.

·      Figure 3 could benefit from highlighting the channel at the hotspot, and adjusting the time range for butterfly plots may enhance clarity.

·      Consider incorporating a figure with panels, letters, and corresponding captions.

·      Were Figures 3 and 4 based on data from a representative subject?

·      It's unclear what type of normalization was implemented in Figure 5. Verify the correctness of the color bar range. The same applies to Figure 6.

·      The topoplots in Figure 7 do not seem to convey clear information. Consider whether data normalization or another tool to present these results may be more suitable.

Discussion:

·      Refocusing lines 343-345 is advisable.

·      A table displaying the mean and standard deviation for each commonly suggested TEP component in M1 would facilitate quantitative comparisons across experimental conditions.

The authors' assertion that "the 'best' hotspot for TEP may differ from that for MEP, regardless of visualization" is thought-provoking. Further investigation into this topic could inspire future research and contribute to a deeper understanding of cortical activation mechanisms.

In conclusion, I commend the authors on their interesting research article and encourage them to explore this topic further in future.

Comments on the Quality of English Language

I would recommend improving language. There are sentences that I suggest to restructure.  

Author Response

The authors of the manuscript titled "Changes in cortical activation by transcranial magnetic stimulation due to coil rotation are not attributable to cranial muscle activation" are conducting a research study investigating a "blind" vs EEG-Informed TMS open loop approach in the motor area. The topic caught my attention, as I find it intriguing that the optimal coil orientation to evoke an MEP may differ from that to evoke a TEP, owing to technological (EEG) or physiological factors (exogenous or endogenous). I hope the authors will delve deeper into this topic in future investigations.

I have some feedback to offer that I believe could assist the authors in enhancing their article.

Abstract:

  • I would suggest revising the term "confound" in line 24 and "TMS-EEG signals" in line 27.

According to the reviewer’s comment, the wording was changed to “decrease the interpretability” and “data obtained”.

  • I found it challenging to follow the distinctions among M1_STANDARD, M1_EMG, and M1_EMG TMS-EEG throughout the paper. Perhaps the authors could use a simpler labeling system, such as Experiment 1, Experiment 2, or a similar approach, defined at the outset and consistently applied throughout the manuscript.

There are only two  experimental conditions in our study (i.e., M1_STANDARD and M1_EMG). We feel that naming the condition with serial numbers, e.g., Exp1 and Exp2, would lead to loss of information resulting in a less readable paper; therefore, we named conditions according to their main experimental features. However we agree that the wording we chose may not be intuitive, so we have used “M1standard“ and “M1emg“ definitions in this new version of the manuscript.

  • I recommend that the authors clearly articulate the problem, the significance of muscle activation and why this has triggered your research question, and emphasize the key findings.

Thanks for the comment. We added the following sentence: “by masking genuine EEG and increasing the reliance on preprocessing methods” (lines 25-26).

Introduction:

  • I would recommend revising the terminology employed in this article. Additionally, careful consideration should be given to the use of "evoke" versus "induce," as well as distinguishing between "muscle artifacts" and "EMG activity."

We thank the reviewer for this comment but we are unsure how to address it. We feel that our use of the mentioned terminology is appropriate in the present context. However, we are happy to revise our work, should the reviewer have concerns about the wording in specific sections of the manuscript.

  • Lines 71-74 are somewhat unclear. Clarifying the statement regarding the drawbacks of monitoring TEP potentials online would be beneficial, as it seems to represent a pivotal aspect of the manuscript's objectives.

We thank the reviewer for this comment. We do not argue against the usefulness of online monitoring of TEPs per se, but rather against free adjustment of coil rotation without considering that cortical neurons and their inputs show preferential activation for a given current direction. Following the reviewers’ comment we have tried to clarify our line of thought (lines 71-74).

  • Bullet points summarizing the contributions would help clarify the authors' focus.

We do not fully understand where these bullet points should be placed and which “contributions” they should summarized, as bullet points are usually not used in scientific papers. We are happy to consider the reviewer’s request if further details are provided.

Experimental Design:

  • The use of noise masking to minimize expectations and TMS sound perception is standard practice in TMS. I'm curious about the effects of continuous noise on cortical excitability. Are there any known effects? There is a recently paper based on MEG study on this topic.

The reviewer is right, a recent paper indeed hinted that continuous white noise can exert excitability modulation effects on the frontal and temporal areas. However, we have no reason to suspect that this is the cause of the differences observed between the two experimental conditions, as the same masking noise was used in both.

  • Were the hotspot and rMT determined with the subject wearing a cap and foam? Please, add this info. 

Yes, both the stimulation hotspot and RMT were found with subjects wearing EEG cap and the described foam layer. This information has been added in lines 109-111.  

More detail regarding about the meaning of "MEPs with the lowest possible stimulation intensity" in line 113 would be helpful.

Thanks for the comment. The sentence the reviewer is referring to is indeed confusing, so we removed it from the text.

Transcranial Magnetic Stimulation and Electromyography:

  • Additional information about the EMG montage used would be beneficial.

We used a belly-tendon montage (added in line 138)

  • Including information about the Inter-Pulse Interval (IPI) utilized would enhance clarity.

The interpulse interval was 2 s with a 10% jitter (added in line 113)

  • Regarding line 156, was the TMS artifact removed and after the EEG signal interpolated?

Signals around the TMS pulse were removed before each round of ICA, to avoid providing artificial signals to the algorhithm; interpolation was performed before filtering, to avoid artefacts due to the presence of zeroes, and at the end of the pipeline, for visualization purposes. The paragraphs has been modified accordingly.

  • A table comparing and detailing the first versus second pipeline would be informative.

Thanks for the suggestion, we added table 1

Statistical Analysis:

  • Did the authors employ any techniques to mitigate the Simpson paradox?·Could a linear mixed model be suitable for conducting multiple comparison statistical analyses on multidimensional data?

We thank the reviewer for this comment. We did not employ any technique to correct for the Simpson paradox, since we feel that it is unlikely to affect our data. Also, we did not use a mixed model because we applied cluster corrected permutation statistics that cannot be integrated in a linear model approach.

  • Clarification regarding the term "first analysis" in line 189 is recommended.

We rephrased for clarity (now line 193-194)

  • What assumptions were made before conducting Canonical Correlation analysis?

We used the concordance correlation coefficient, rather than canonical correlation. Also, we employed permutation statistics, which are robust to outliers and deviation from normality.

  • The meaning of "extreme" corrected permutation approach in line 207 requires clarification.

The explanation of the extreme correction is present in the text, in lines 219-222

  • Did the authors apply baseline correction before conducting TF analysis?

Yes, we did. This information can be found in lines 232-233

Results:

  • Table 1 is insightful. Including the formulas used to calculate the mean and standard deviation, it is something I would recommend. Additionally, a more informative caption is suggested. The same applies to Table 2.

We added a sentence in the captions of table 1 and 2 (now table 2 and 3) to make them more informative. We feel that formulas for calculating mean and standard deviation and common knowledge enough; therefore, we would rather avoid writing them not to decrease the readability of the paper unnecessarily.

  • Figure 1 would benefit from including units on the y-axes. Consider presenting a plot with shaded ranges representing standard deviation to illustrate the overlap of muscle artifact between approaches.

We thank the reviewer for this comment. We added units on X and Y axis. Regarding the standard deviations, the reviewer states a good point. We prefer to show individual EMG traces to give the reader a thorough account of individual variability of cranial muscle activation following TMS. However, following the reviewer’s remark, we have added a formal comparison of muscle artifact from Fc5 between conditions, by averaging the signal within 5 and 25 ms from TMS pulse. Hence, the text (188-190, 263-264) was modified accordingly, and a boxplot was added to figure 1.

  • How many ICA components were rejected?

The information has been added in lines 287-288

  • A group-level topoplot for each condition would provide insight into TEP characteristics.

Thanks for the comment. Figures 3 and 4 provide group level topoplots for each condition.

  • Figures 2 and 3 raise concerns about the range of butterfly plots magnitude relative to the expected motor area range.

We thank the reviewer for spotting this. There was indeed a mistake on the y axis of the upper plot in figure 3, which has been corrected.

  • Investigating MEP amplitude variability between different approaches could yield valuable insights for further statistical analysis.

Thanks for the comment. As a matter of fact, since the stimulation we applied was subthreshold, no MEPs were actually recorded from the patients.

  • Figure 3 could benefit from highlighting the channel at the hotspot, and adjusting the time range for butterfly plots may enhance clarity.

The x axis already contains all the time windows used in the analysis and gives the reader an idea of signals with slightly longer latencies; therefore, we feel that changing it would result in loss of significant information. Also, we added a red star to indicate the stimulation point as suggested.

  • Consider incorporating a figure with panels, letters, and corresponding captions.

Thanks for the suggestions. We modified the figures as suggested.

  • Were Figures 3 and 4 based on data from a representative subject?

No, the figures represent group averages.

  • It’s unclear what type of normalization was implemented in Figure 5. Verify the correctness of the color bar range. The same applies to Figure 6.

The figures illustrate the methodology applied to identify time-windows of interest, that were later used in the analysis. TRSP values were first thresholded to baseline (-500 to -200 ms from TMS pulse) and then z scored with respect to the whole time by frequency image to find the time of interest. ITPC underwent the same processing, but was not normalized with respect to baseline, since this would have limited the interpretability of the variable (which ranges from -1 to 1).

  • The topoplots in Figure 7 do not seem to convey clear information. Consider whether data normalization or another tool to present these results may be more suitable.

Thanks for the comment. The authors feel that the image is quite informative, representing the variation between sessions in power within predefined frequency of interest and data drive time-windows. Normalization has been performed with respect to baseline, as specified in the text and in the previous answer.

Discussion:

  • Refocusing lines 343-345 is advisable.

Thanks for the comment. We modified the sentence (now line 359-360)

  • A table displaying the mean and standard deviation for each commonly suggested TEP component in M1 would facilitate quantitative comparisons across experimental conditions.

Thanks for the suggestion. Although it would be useful to display our results in a table, this is actually impractical, since we utlilized a multivariate approach that takes into account the variation of signal across all scalp electrodes. Hence, we cannot represent the results as means and standard deviations, but we identified the significantly different clusters in figure 3 and 4.

The authors' assertion that "the 'best' hotspot for TEP may differ from that for MEP, regardless of visualization" is thought-provoking. Further investigation into this topic could inspire future research and contribute to a deeper understanding of cortical activation mechanisms.

In conclusion, I commend the authors on their interesting research article and encourage them to explore this topic further in future.

We thank the reviewer for this kind comment.

Please see the attached document for the updated version of the article

Round 2

Reviewer 2 Report

Comments and Suggestions for Authors

Dear authors,

I still have some concerns regarding the figures presented in your manuscript. In Figure 1(a), there are discrepancies observed between the representation of M1sta (red line in the first plot) and M1emg (blue line in the second plot). Specifically, the M1sta in the first plot appears to differ from that in plot three, not only in terms of their y-axes scale but also in magnitude. 

I recommend aligning the legend in Figure 2 and others with the formatting used for M1sta and M1emg along the manuscript.

Moving on to Figure 3, I'm curious about whether the color bar used for DIFF maintains the same range for both M1sta and M1emg. Upon inspection, I noticed some discrepancies that raise suspicion.

It's possible that the black asterisks mentioned in the caption are not clearly visible.

Furthermore, while I appreciate the authors' response regarding the 2uV amplitude of the average butterfly plots, I still harbor some reservations. However, I acknowledge and understand the explanation provided.

I want to commend you on the thoroughness of your revision. It's evident that you put in a lot of effort, and it's greatly appreciated.

Author Response

Dear authors,

I still have some concerns regarding the figures presented in your manuscript. In Figure 1(a), there are discrepancies observed between the representation of M1sta (red line in the first plot) and M1emg (blue line in the second plot). Specifically, the M1sta in the first plot appears to differ from that in plot three, not only in terms of their y-axes scale but also in magnitude. 

- Thanks for noticing this, indeed there has been a mistake in Y axis labelling, which has now been corrected.

I recommend aligning the legend in Figure 2 and others with the formatting used for M1sta and M1emg along the manuscript.

- Thanks for the suggestion. We have now adopted the same nomenclature all throughout the figures-

Moving on to Figure 3, I'm curious about whether the color bar used for DIFF maintains the same range for both M1sta and M1emg. Upon inspection, I noticed some discrepancies that raise suspicion.

- Yes, that is the case: the colorbar is the same across M1sta, M1emg and M1diff, so all data are represented on the same scale. 

It's possible that the black asterisks mentioned in the caption are not clearly visible.

- The asterisks representing the significant electrodes are indeed visible only on the third row, where significant differences between conditions are highlighted. This is a relatively standard approach for highlighting significant electrodes, wo we feel that they should not be changed.

Furthermore, while I appreciate the authors' response regarding the 2uV amplitude of the average butterfly plots, I still harbor some reservations. However, I acknowledge and understand the explanation provided.

I want to commend you on the thoroughness of your revision. It's evident that you put in a lot of effort, and it's greatly appreciated.

Thanks for the comment